# Panoramic Research on *Sesbania cannabina*: Germplasm Resources, Phytochemical Constituents, Biological Activities, and Applications

**DOI:** 10.3390/ijms262412129

**Published:** 2025-12-17

**Authors:** Ni Zhang, Junhua Meng, Qianqian Wang, Cuiping Liu, Duanrui Cao, Keping Hu, Xiaofeng Cao, Huanwen Chen

**Affiliations:** 1School of Pharmacy, Jiangxi University of Chinese Medicine, Nanchang 330004, China; zhangni61116@163.com (N.Z.); mengjunhua0131@163.com (J.M.); shishiruyi2002@163.com (Q.W.); gg20251024@163.com (C.L.); jxcdr0121@163.com (D.C.); 2The Jiangxi Province Key Laboratory for Diagnosis, Treatment, and Rehabilitation of Cancer in Chinese Medicine, Jiangxi University of Chinese Medicine, Nanchang 330004, China; 3Institute of Medicinal Plant Development, Chinese Academy of Medical Sciences & Peking Union Medical College, Beijing 100193, China; hkpucla@163.com; 4Institute of Genetics and Developmental Biology, Chinese Academy of Sciences, Beijing 100101, China

**Keywords:** *Sesbania cannabina*, germplasm resources, chemical constituents, pharmacological activities

## Abstract

*Sesbania cannabina* (Retz.) Poir. (*S. cannabina*), as an important leguminous plant in the genus *Sesbania*, plays a crucial role in agricultural sustainability and ecological restoration. This review systematically summarizes its current research status and conducts a comprehensive analysis of germplasm resources, chemical composition, biological activity and multi-field applications. Regarding germplasm resources, this article reviews the germplasm resource characteristics of *Sesbania*, which is widely distributed and has strong stress resistance, as well as its huge application potential in agriculture, ecological restoration and other fields. At the chemical composition level, this study reviews the chemical composition of various parts of *Sesbania*, with a focus on analyzing the dynamic change patterns of its rich secondary metabolites, and summarizes the related extraction, separation, and analytical identification methods. Regarding medicinal value and bioactivity, this article reviews the traditional medicinal value of *S. cannabina*, with a focus on exploring the mechanism of action and safety of its modern pharmacological activities such as antioxidation, anti-inflammation, and immunomodulation. In conclusion, the comprehensive value of *S. cannabina* is remarkable. Future research should delve deeply into its resources, clarify the active mechanism and develop high-value applications to fully tap into its multi-functional potential.

## 1. Introduction

*S. cannabina* is an annual herb or subshrub of the Fabaceae family, widely distributed throughout tropical and subtropical regions worldwide, particularly in the coastal and southern provinces of China. With its tall growth habit and well-developed root system, this species demonstrates remarkable adaptability to diverse environmental conditions, including saline-alkali soils, waterlogging, and nutrient-poor habitats. Owing to these traits, *S. cannabina* plays an important role in both agriculture and ecological restoration. As a traditional green manure crop, *S. cannabina* can improve soil fertility through nitrogen fixation, with a protein content as high as 25%. Additionally, under comparable water quality conditions, the dry matter yield of *S. cannabina* can reach up to 45 tons per hectare annually, significantly higher than that of “queen of forages” alfalfa, making it a valuable source of livestock feed [1]. In ecological restoration, *S. cannabina* is a preferred crop that can promote vegetation restoration in degraded ecosystems. The PH value of the soil improved by it decreased from 8.0 to 7.5. In addition, the total nitrogen content in the improved soil rose from 0.816 to 1.25 [2]. Previous research employing genomic and molecular approaches has clarified the adaptive mechanisms of *S. cannabina* and laid out potential genetic-improvement routes to optimize cultivars [3,4]. Recent studies have reported a wide array of secondary metabolites and bioactive compounds in *S. cannabina*, with multiple constituents demonstrating notable pharmacological activities [5]. For example, galactomannan (GM) extracted from *S. cannabina* seeds has been shown in in vitro studies to exhibit an inhibition rate of 93.79% against Michigan Cancer Foundation-7 (MCF-7) cells at a concentration of 400 µg/mL [6].

Despite the broad application potential of *S. cannabina* agricultural production, ecological restoration, and pharmaceutical development, significant knowledge gaps remain. In particular, the germplasm resource evaluation system is still underdeveloped, and the establishment of a core collection is urgently needed. The spatiotemporal distribution and accumulation patterns of chemical components are not fully understood, and efficient extraction techniques require further optimization. Moreover, studies on pharmacological mechanisms are largely confined to in vitro models, lacking clinical evidence and precise target validation. Furthermore, existing research largely limits its analysis of *S. cannabina* to specific disciplinary perspectives, lacking a systematic evaluation from a multidisciplinary approach. These fragmented insights make it difficult to form a holistic understanding of its multifunctional value, thereby hindering the comprehensive development and utilization of *S. cannabina*.

This review systematically integrates research progress on *S. cannabina* from three dimensions: germplasm resources, chemical constituents, and pharmacological effects. It offers a comprehensive perspective on this multi-functional plant. In terms of germplasm resources, we summarize its geographical distribution and ecological adaptability, with emphasis on molecular mechanisms underlying salt-alkali and heavy metal tolerance. Recent advances in germplasm conservation and breeding of improved varieties are also summarized, providing insights for future molecular breeding. Regarding chemical constituents, this review analyzes and compares the chemical profiles of different plant parts, highlighting tissue-specific accumulation of primary and secondary metabolites. In the area of pharmacological effects, we bridge traditional uses with modern pharmacological studies, summarizing its antioxidant, immunomodulatory, and antitumor activities, as well as emerging roles in neuroprotection and gut microbiota regulation. This review, through a unique interdisciplinary integration, establishes a comprehensive framework connecting germplasm resources, chemical composition, and pharmacological functions. It aims to provide a theoretical foundation for both traditional and modern applications of *S. cannabina*, while offering a thorough scientific basis for its future comprehensive development and utilization.

This paper presents a comprehensive review of the germplasm resources, chemical composition, and pharmacological effects of *S cannabina*. To ensure comprehensive coverage, we queried multiple domestic and international databases, including Web of Science, PubMed, China National Knowledge Infrastructure (CNKI), ancient Chinese classics collections, and news reports. During the search, specific keyword combinations such as “*Sesbania* germplasm resources,” “*Sesbania* ecological restoration,” “*Sesbania* genetics,” “*Sesbania* chemical composition”, and “*Sesbania* pharmacological effects” were used. These keywords were selected to encompass studies related to the utilization of *Sesbania* germplasm resources, its chemical composition, and pharmacological effects. Given the need for a comprehensive review of the multi-faceted research on *Sesbania*, the sources included in the reference list were not limited to English-language studies but also included ancient Chinese classic collections and news reports. Furthermore, due to the limited availability of relevant materials, no specific time frame was set for the references included in this review. The selection criteria were designed to ensure the inclusion of more relevant literature while maintaining the reliability of the sources. The initial search results underwent a systematic screening process, conducted independently by all authors. This method ensured impartiality in the selection process and removed duplicate references. After the preliminary screening, 100 references were included in the review. Finally, a thorough and systematic analysis of these references was conducted, providing a detailed discussion on the germplasm resources, chemical composition, and pharmacological effects of *Sesbania*. This transparent approach facilitates a focused yet comprehensive synthesis of the existing literature.

## 2. Germplasm Resources

### 2.1. Distribution and Ecological Adaptability

#### 2.1.1. Global and China Geographic Distribution

*Sesbania* comprises approximately 50 species worldwide, mainly distributed in tropical and subtropical regions, particularly in South and Southeast Asia, including India, Indochina, and Malaysia. In China, *Sesbania* is concentrated in coastal provinces such as Fujian, Guangdong, Jiangsu, and Hainan, and its range has progressively extended northward into North and Northeast China. Within China, the primary species include *S. cannabina* [7], *Sesbania bispinosa* (*S. bispinosa*) [8], *Sesbania sesban* (*S. sesban*) [9], *Sesbania javanica* (*S. javanica*) [10], and *Sesbania grandiflora* (*S. grandiflora*) [11]. *S. cannabina* is the most widely cultivated species in China, characterized by its dense foliage, substantial biomass, extensive root system, and numerous large root nodules. The differences among *Sesbania* species are shown in Figure 1 and Table 1.

#### 2.1.2. Adaptability to Abiotic Stresses Such as Salinity, Alkalinity, and Heavy Metals

*S. cannabina* exhibits strong tolerance to multiple abiotic stresses, including drought, salinity, prolonged flooding, and nutrient-poor soils. Its typical habitats encompass coastal marshes, saline-alkali wastelands, reservoir drawdown zones, and heavy metal-contaminated tailings areas. The species tolerates a wide soil pH range and performs particularly well under saline-alkali and waterlogged conditions; reported thresholds for normal growth include soil salt concentrations up to 6 g/kg and alkaline soils with pH values around 9.5. These characteristics underpin its potential for land reclamation and phytoremediation of degraded habitats [13,14,15]. Additionally, *S. cannabina* has strong tolerance to heavy metal pollution. Experimental studies show that its roots and root nodules accumulate cadmium, lead, zinc, and other metals at markedly higher concentrations than stems and leaves, with tissue metal levels positively correlated with soil metal concentrations. Such accumulation patterns indicate the species’ potential utility for phytoremediation of contaminated sites [16,17].

#### 2.1.3. Cultivation History and Current Status

*S. cannabina*’s cultivation history can be traced back to ancient China; as early as the Western Han Dynasty, Sima Xiangru mentioned it in *Shanglin Fu*: “feeding on jing grass and algae, chewing on water caltrop and lotus root”, which is considered the earliest recorded reference to *S. cannabina* in ancient literature. Traditionally, *S. cannabina* has been widely used as green manure, for medicinal purposes, and as a raw material for hemp production. Its modern large-scale cultivation commenced in the 1950s. *S. cannabina* was designated a pioneer crop in the coastal saline-alkali regions of Jiangsu and Shandong owing to its tolerance of salinity and waterlogging. Standardized cultivation practices include warm-water seed soaking to break the waxy seed coat, shallow sowing (≤2 cm), and combined application of phosphorus and potassium fertilizers. The current cultivation areas have since expanded from coastal provinces to the Yellow River Delta, Heilongjiang Province, and heavily saline-alkali regions such as Aksu (Xinjiang), where *S. cannabina* is employed for saline-alkali land remediation [18,19,20]. In recent years, “Lu Jing No. 5” and “Zhongkejing No. 1” have become the main cultivated varieties of *S. cannabina* in China. In terms of cultivation practices, the Chinese Academy of Sciences has developed an integrated agricultural system in Aksu, Xinjiang, that combines soil improvement, forage production, and crop-livestock integration through *Sesbania* cultivation. This system achieves fresh forage yields of 4.5–7.5 kg/m^2^, with crude protein content reaching 15–20%, providing an innovative approach to alleviating land resource constraints [21]. At the same time, an intercropping system consisting of one row of sweet sorghum with three rows of *S. cannabina* has been promoted in the saline-alkali soils of Xinjiang. This configuration takes advantage of *S. cannabina*’s nitrogen-fixing ability to enrich soil nitrogen, while the deep root system of sweet sorghum helps reduce salt leaching, thereby synergistically improving land-use efficiency [22].

### 2.2. Germplasm Types and Genetic Diversity

#### 2.2.1. Ecological Type/Varietal Characteristics in Different Regions

*S. cannabina* has undergone significant ecological diversification, forming multiple ecotypes across tropical to temperate regions globally. Its adaptive evolution is jointly driven by geographic isolation and environmental selection pressures. In the northeastern soda-alkali regions of China, the novel cultivar “Zhongkejing No. 1” (developed by Academician Cao Xiaofeng’s team) demonstrates remarkable agronomic traits, including a plant height of up to 2.8 m and a fresh forage yield of 3.04 kg/m^2^. This variety thrives under severe saline-alkaline conditions (soil pH > 9.5) and significantly reduces soil pH by approximately 0.5 units [23]. The “Lu Jing” series (Lu Jing No. 1, No. 2, and No. 5), developed by the Shandong Academy of Agricultural Sciences, exhibits robust salt tolerance in the saline-alkali soils of the Huang-Huai-Hai area, exemplified by regions such as Dongying, Shandong. This variety exhibits a growth period of 130–170 days, producing fresh forage 5.55–6.6 kg/m^2^, with its dry hay containing 18.4% crude protein. Owing to these properties, it has multifunctional applications as animal forage, green manure, and a raw material for gelatin extraction [24]. A distinctive characteristic of the stem-nodulating *Sesbania* found in coastal Jiangsu is the formation of nitrogen-fixing nodules on its stems. With 890 to 1650 stem nodules per plant, its nitrogen-fixation capacity is doubled compared to *S. cannabina* varieties. This enhanced capability renders it particularly suitable for the reclamation of tidal flats [25]. These ecological adaptations exemplify the evolutionary response of *Sesbania* to regional climatic conditions, soil salinity, and agricultural practices, thereby underpinning targeted breeding initiatives and regionalized cultivation.

#### 2.2.2. Morphological Differences

Plants within the genus *Sesbania* exhibit a range of growth forms and can be broadly categorized as herbaceous, shrubby, or small trees. The primary morphological distinctions among these types are observed in plant height, leaf architecture, flower size, and pod characteristics. The herbaceous type includes *S. cannabina* and *S. javanica*. The former plant is 2 to 4 m tall. Its leaves are uniformly pinnate, with 10 to 30 pairs of linear-lanceolate leaflets on them. The pods are slender, cylindrical or nearly beaded, measuring 15 to 25 cm in length. The latter plant is 1 to 3 m tall, has 20 to 30 pairs of small leaves, and its fruit pods are 12 to 22 cm long. The shrubby type is represented by *S. sesban* and *S. bispinosa*. The former is a shrub, 2 to 4 m tall, with 10 to 20 pairs of small leaves, and its fruit pods are 15 to 23 cm long. The latter is 1 to 3 m tall, with 20 to 40 pairs of small leaves and a fruit pod length of 15 to 22 cm. The small tree type is represented by *S. grandiflora*, which can grow to a height of 4 to 10 m. Its fruit pods are large, measuring 20 to 60 cm in length, and its seeds are elliptical [12]. The differences in *Sesbania* in the above aspects may be one of the main reasons that directly affect its utilization of germplasm resources and the differences in active components in different parts.

#### 2.2.3. Genetic Diversity Research

Structural genomic variations are a major driver of molecular-level diversity in *Sesbania*. A notable example is a 27 Mb pericentromeric inversion on chromosome 5 of *S. bispinosa*, which suppresses recombination by approximately 90%. This inversion has facilitated the fixation of a salt-tolerant haplotype harboring a cluster of phosphate transporter (*PHT1*) genes. Complementary to these findings, a genome-wide association study (GWAS) identified six loci associated with salt tolerance. Among these, the anthocyanin synthase gene (*SbANS*), located within a region of 42.71–42.90 Mb on chromosome 5, was strongly associated with the trait. Genomic analyses have revealed that the common species *S. cannabina* is an allotetraploid (2n = 4x = 24). Its subgenome differentiation has been influenced by transposable element (TE) expansion. The B subgenome, which diverged earlier (approximately 7.8 million years ago), shows closer affinity to *Sesbania rostrata*. In contrast, the A subgenome diverged more recently (about 1.5 million years ago) and is characterized by a significant enrichment of SIRE transposable elements (52.8%). This TE expansion is associated with a 3–5-fold upregulation of PHT1gene expression under alkaline stress conditions [26,27].

Genetic diversity in *S. cannabina*, a pioneer species for saline-alkali land improvement, is studied primarily in two dimensions: the plant’s intrinsic mechanisms of stress response and its symbiotic interactions with microorganisms. Under salt stress, *Sesbania* mounts a response through temporal reprogramming of root gene expression. This response included 571 and 1765 differentially expressed genes (DEGs) identified at 3 h and 27 h, respectively, which were notably enriched in pathways of secondary metabolism (e.g., phenylpropanoid biosynthesis), carbon metabolism, and glycolysis. Transcription factors (TFs), spanning 26 families such as MYB, bHLH, AP2/EREBP, and NAC, function as central regulators. Notably, 66 TFs exhibited bidirectional expression divergence over the 27 h stress period, revealing a polygenic cooperative network underlying salt tolerance. A novel rhizobial strain, *Rhizobium Sesbaniae* ZK1^T^, was isolated from the stress-tolerant cultivar *Sesbania* “Zhongkejing No. 1”. Phylogenetic analysis based on the whole genome and 16S rRNA gene sequences confirmed its status as a new species within the genus Rhizobium. The strain carries essential symbiotic genes (nodABC-nifHcluster), tolerates extreme conditions (2.0% NaCl, pH 4.0–10.0), and demonstrates phosphate-solubilizing capacity. Pot experiments verified that inoculation significantly enhances host biomass and nodulation efficiency. Furthermore, it exhibited a broad-spectrum growth-promoting effect on polyploid *Sesbania* cultivars, underscoring a conserved host-microbe coevolutionary mechanism [28,29].

The genetic diversity of *S. cannabina* is shaped by both geographical isolation and environmental selection. Coastal ecotypes have fixed salt-tolerant haplotypes through chromosomal inversions and are characterized by high biomass, deep root systems, and efficient phosphorus uptake. In contrast, inland ecotypes show higher genetic diversity but lower salt tolerance. Subgenome differentiation in the allotetraploid and the expansion of transposable elements are considered major drivers of adaptive evolution. Future research should integrate multi-omics approaches to identify stress-tolerance genes and advance the use of marker-assisted breeding for saline-alkali land improvement.

### 2.3. Collection and Conservation of Germplasm Resources

#### 2.3.1. Current Status of Germplasm Repositories at Home and Abroad

Internationally, germplasm conservation of *Sesbania* is primarily undertaken by major international agricultural research institutions and regional gene banks. Research indicates that seeds from different *Sesbania* germplasm possess exceptional viability retention capabilities under suitable conditions. Seeds of *Sesbania tomentosa* from Hawaii demonstrate exceptional longevity, maintaining a germination rate of up to 88.9% (range: 73–100%) after approximately 30 years of storage under natural conditions. This evidence underscores the feasibility of long-term conservation for endangered *Sesbania* germplasm [30]. The collection and conservation of *Sesbania* germplasm by international research institutions is driven by both agricultural and biodiversity imperatives, focusing notably on securing the genetic resources of endangered wild species.

Systematic collection and preservation of *Sesbania* germplasm in China has established a substantial resource base. By the late 1980s, 56 distinct varieties had been assembled through collection and introduction. Following consolidation and classification, 47 accessions were successfully preserved, the majority of which belong to *S. cannabina* [31]. The Shandong Provincial Crop Germplasm Resource Center has systematically collected *Sesbania* germplasm resources from the coastal areas of Shandong Province. Through field surveys of 132 villages across 82 townships, 903 foundational samples were collected, and 224 endangered local and wild resources were rescued and preserved. Among these, 75 drought-tolerant and salt-tolerant *Sesbania* germplasm resources were selected [32]. Led by Academician Xiaofeng Cao, the saline-alkali land improvement team at the Institute of Genetics and Developmental Biology, Chinese Academy of Sciences, has made significant advances in *Sesbania* research. Their work has established it as a saline-alkali-tolerant pioneer species, elucidated its genome and stress-resistance mechanisms, developed China’s first genetic transformation system for the genus, and bred elite cultivars such as “Zhongkejing No. 1”, known for their high yield, quality, and stress tolerance [33]. These efforts have substantially augmented the national *Sesbania* germplasm repository, serving as a key genetic resource for the improvement of saline-alkali lands.

#### 2.3.2. Preservation Methods and Breeding Techniques

The conservation of *S. cannabina* germplasm employs two principal strategies: in situ and ex situ conservation. Ex situ methods primarily involve seed storage at low temperatures, whereas in situ conservation aims to protect natural populations within their native habitats. Recently, advancements in biotechnology have facilitated the application of sophisticated preservation and propagation techniques to *Sesbania* germplasm.

To address critical challenges in the safe storage of germplasm, namely low-temperature refrigeration with dehumidification and seed drying, the Shandong Provincial Crop Germplasm Resource Center developed proprietary repository management software and a suite of pre-processing equipment. This initiative scientifically established optimized storage parameters: a repository temperature of (−4 ± 2) °C, relative humidity ≤ 45%, germplasm purity ≥ 99%, germination rate ≥ 90%, and a seed moisture content of 6% ± 2%. These parameters are critical for the long-term preservation of *Sesbania* germplasm [32].

In *Sesbania* breeding, traditional methods like mass selection and single-plant selection are routinely employed. By applying these techniques to 17 wild mung bean resources collected from the Shandong coast and targeting high yield, high protein content, and stress tolerance, the Shandong Provincial Crop Germplasm Resource Center developed the lines “Lu Jing No. 1” and “Lu Jing No. 3” [32].

Breakthrough progress has also been achieved in the application of modern biotechnology in the breeding of *Sesbania*. In 2023, researchers successfully established a genetic transformation method for *Sesbania*, filling a gap in the genetic transformation system for this crop. This method utilizes the cotyledons or hypocotyls of *Sesbania* as starting explants to establish a rapid and efficient regeneration system. Glyphosate herbicide is employed as a screening agent to obtain transgenic positive lines [34].

#### 2.3.3. Progress in Introduction and Domestication

*S. cannabina*, a highly stress-tolerant leguminous plant, has made remarkable progress in recent years in terms of introduction and domestication, particularly demonstrating great potential in saline-alkali land improvement and forage resource development. The research team led by Academician Cao Xiaofeng systematically collected more than 400 wild *S. cannabina* germplasm accessions from diverse regions in China and abroad. Through approaches including wild domestication, hybrid breeding, and gene editing, the team developed a new salt-tolerant and high-yielding cultivar, “Zhongke Jing No. 1”. This cultivar entered the national grass regional trials in 2023. In demonstration plantings on highly soda-alkaline soils (pH > 9) in the Northeast Songnen Plain, it achieved a first-cut fresh forage yield of 3.04 kg/m^2^. Moreover, via root nitrogen fixation and straw incorporation, the cultivar reduced soil pH by 0.5 units and increased soil organic matter content by more than 10% [35,36]. To address the constraints of small seed size and poor soil penetration in *Sesbania*, the research team developed a suite of tailored cultivation techniques during the domestication process. These included the use of shallow, precision seed drills, the application of selective herbicides, and integrated water-fertilizer management. This approach effectively improved seedling establishment, enabling the achievement of a germination rate exceeding 90% [37]. Genomic studies have revealed the molecular mechanisms underlying stress tolerance in *S. cannabina*. Its allotetraploid genome (approximately 2.087 Gb) contains more than 90,000 genes, with specific phosphate transporter PHT1 family genes being induced under saline-alkaline stress to alleviate phosphorus deficiency. In addition, CRISPR-Cas9-mediated knockout of genes controlling internode length and leaf abscission has further optimized plant architecture, lodging resistance, and protein retention capacity [27].

### 2.4. Current Status and Potential of Germplasm Utilization

#### 2.4.1. Agricultural Utilization

*S. cannabina* is a leguminous green manure crop of high agricultural value. It yields up to 45 tons of fresh weight per hectare annually and demonstrates strong tolerance to salinity-alkalinity, waterlogging, and drought. Furthermore, *S. cannabina* is characterized by a unique dual nitrogen-fixing system, involving both root and stem nodules. This system confers a nitrogen fixation efficiency significantly surpassing that of soybean and alfalfa, underscoring its considerable value as a multifunctional green manure [38,39]. The intercropping of *Sesbania* in tea plantation systems, combined with a 30% reduction in chemical fertilizer, enhanced soil health and tea quality. This practice increased soil organic carbon, stimulated the activity of acid phosphatase and protease, enriched bacterial diversity, and improved tea quality indicators such as amino acid and water-soluble extract content [40].

*S. cannabina* ameliorates saline-alkali soils by reducing surface salt content, pH, and conductivity during growth, while returning substantial nutrients to the soil when incorporated as green manure. This process enhances organic matter, total nitrogen, and readily available nutrient content, thereby significantly boosting the yield and quality of subsequent crops such as silage corn [41]. Furthermore, *S. cannabina* demonstrates allelopathic potential by producing and releasing specific compounds that suppress weed growth. Concurrently, these compounds stimulate soil microbial and enzymatic activity, contributing to improved soil fertility [39].

Although *S. cannabina* is a feed resource with high crude protein content, its ensiling is often hindered by low concentrations of water-soluble carbohydrates (WSC < 5% dry matter) and a high buffering capacity [38]. Research indicates that the silage quality of *S. cannabina* is significantly improved by mixing it with maize or sweet sorghum. When combined with inoculation by compound lactic acid bacteria or cellulase treatment, this approach enhances fermentation, aerobic stability, and fiber conversion, while suppressing harmful microorganisms. Consequently, crude protein retention and overall feed nutritional value are increased [42,43]. This mixed silage strategy offers a practical approach for the dual-purpose application of *S. cannabina* as both green manure and forage in saline-alkali agricultural systems.

#### 2.4.2. Ecological Restoration

*S. cannabina* demonstrates significant potential in soil ecological restoration. It contributes to saline-alkali soil amelioration through biological desalination, which effectively lowers salinity, moderately reduces pH, and concurrently enhances soil organic matter and alkali-hydrolyzable nitrogen content [44]. The cultivation of *S. cannabina* modifies the soil ionic profile by reducing Na^+^ and Cl^−^ concentrations and promoting Ca^2+^ accumulation, which in turn facilitates the leaching of sodium [45]. In addition, *S. cannabina* forms symbiotic associations with rhizobia and arbuscular mycorrhizal fungi. These symbioses enhance plant growth under stress and contribute to improved soil fertility [46]. This tripartite symbiosis enhances the remediation of polycyclic aromatic hydrocarbons (PAHs) in soils through rhizosphere processes that stimulate the release of water-soluble phenolic compounds, enrich PAH-degrading bacterial populations, and boost the activity of enzymes like polyphenol oxidase. These effects collectively accelerate PAH degradation. Under saline-alkaline conditions, the synergistic action of rhizobial nitrogen fixation and arbuscular mycorrhizal fungal association further improves nitrogen use efficiency and strengthens host plant salt tolerance [47].

The integrated restoration strategy further enhances the soil improvement effects of *S. cannabina*. For example, the co-application of biochar and effective microorganisms (EM) synergistically elevates soil nutrient content (e.g., total nitrogen, available phosphorus, and available potassium), organic carbon pools, and microbial biomass carbon. This improvement in soil properties thereby promotes the growth of *S. cannabina* and accelerates the ecological restoration of saline-alkali land [48]. In arsenic-contaminated soils, intercropping with *Pteris vittata* enhances arsenic phytoextraction efficiency by 19.3–53.9%. This system also increases arsenic bioavailability via rhizosphere acidification induced by nitrogen fixation and root exudation, without compromising agricultural product safety [49].

Long-term cultivation of *S. cannabina* can significantly improve microbial community structure and function. A three-year field study confirmed that *S. cannabina* cultivation reduces soil electrical conductivity (EC) while increasing the concentrations of total carbon, total nitrogen, and nitrate nitrogen. It also enriched beneficial diazotrophs and fostered a more stable, complex microbial co-occurrence network, which enhanced the soil’s carbon and nitrogen cycling functions [50]. Additionally, *S. cannabina* harbors beneficial rhizobacteria, such as *Enterobacter* sp. *N102*, with functionalities that include nitrogen fixation, phosphorus solubilization, potassium mobilization, and phytohormone production. These bacteria sustain growth-promoting effects under high salinity, highlighting the potential of using this plant-microbe system in the bioremediation of saline-alkali soils [51].

Owing to its unique physiological adaptability, efficient microbial partnerships, and utility in combined remediation systems, *S. cannabina* significantly enhances the physicochemical properties and ecological functions of degraded saline-alkali soils, thereby establishing it as an economically and environmentally sustainable restoration strategy. Future integration of advanced monitoring techniques could further enable precision remediation approaches using this plant [52].

## 3. Chemical Composition

### 3.1. Primary Metabolite Components

#### 3.1.1. Choline

Choline serves as a fundamental component of cell membranes (phospholipids) and regulates growth and development for plants [53], and it functions as a precursor to neurotransmitters and a methyl donor, possessing medicinal properties for humans [54]. *S. cannabina* contains choline. Huanwen Chen and his team rapidly detected choline (2-Hydroxy-N, N, N-trimethylethan-1-aminium) in *S. cannabina* using internal extractive electrospray ionization mass spectrometry (iEESI-MS). iEESI-MS is a technique that requires no sample pretreatment and preserves *S. cannabina*‘s real composition [55].

#### 3.1.2. Polysaccharide

Polysaccharides serve as primary metabolites that form cell walls (such as cellulose) to provide structural support, and function as energy storage substances (such as starch) to sustain vital activities for plants themselves [56,57]. Polysaccharides also exhibit pharmacological activity. Specific bioactive polysaccharides demonstrate therapeutic value through multiple pathways—including immune modulation and antioxidant effects—to enhance bodily functions and prevent or treat diseases [58,59]. *Sesbania* gum (SG), a natural polysaccharide extracted from the endosperm of *S. cannabina* seeds, is primarily composed of GM—a polymer of D-galactose and D-mannose [60]. Additionally, other polysaccharides such as xylan and arabinan have been identified in *S. cannabina* [6].

#### 3.1.3. Adenosine

*S. cannabina* contains adenosine, particularly purine derivatives (6-Armino-9-β-D-ribofuranosyl purine) isolated from byproducts obtained after seed refining (Figure 2a) [61]. That represents the first isolation of this compound from the *Sesbania* genus. Adenosine serves as a fundamental substance for the life activities of plant cells and also constitutes an important class of pharmacologically active compounds [62,63].

### 3.2. Secondary Metabolite Components

#### 3.2.1. Pyranoids

Research has found that *S. cannabina* contains Pyranoids. Pyranone alkaloids (2-Hydroxy-3-methyl-γ-pyrone) isolated from byproducts obtained after seed refining (Figure 2b) [61].

#### 3.2.2. Phenylbutazone

Phytochemical investigation of the stems of *S. cannabina* has led to the identification of various 2-arylbenzofuran derivatives. These include sesbcanfuran A (Figure 2c), sesbcanfuran B (Figure 2d), sesbagrandiflorain E (Figure 2e), 2-(4-hydroxy-2-methoxyphenyl)-6-methoxy-3-benzuofur ancarboxylic acid methyl ester (Figure 2f), spinosan A (Figure 2g), 2-(2′-methoxy, 4′-hydroxy)-aryl-3-methyl-6-hydroxy-benzuofuran (Figure 2h), sesbagrandiflorain A(Figure 2i), and sesbagrandiflorain B (Figure 2j) [64].

#### 3.2.3. Flavonoids

Phytochemical analysis confirmed the presence of flavonoids in *S. cannabina*, with a quantified content of 0.40%. Specific colorimetric tests showed strong positive reactions with 1% aluminum nitrate and 1% FeCl_3_, but negative reactions with the hydrochloric acid-zinc powder and boric acid tests. These results indicate that the flavonoids present are likely aurone and catechins, without 5-hydroxyflavones or 2-hydroxychalcone structures [65].

#### 3.2.4. Saponins

*S. cannabina* contained saponin compounds at a concentration of 2.28%. The Liebermann-Burchard test yielded a positive reaction (final color turning reddish-purple). The froth test indicated the presence of triterpene saponins [65].

#### 3.2.5. Other Ingredients

Reports indicate that the content of phenolic compounds and tannins in *S. cannabina* was 1.86%, with organic acids, coumarins, and terpenoid lactones also present. Chemical testing indicated positive results for both phenolic compounds and tannins. Tests for coumarins and terpenoid lactones also yielded positive outcomes [65].

### 3.3. Differences in Composition Among Different Plant Parts

The composition of different parts of *S. cannabina* differs significantly. The roots are rich in fibrous substances, providing mechanical support. The cuticle waxes of stems and leaves are primarily composed of primary alcohols, with leaves exhibiting a significantly higher proportion than stems. However, only stems contain diol compounds and a series of 2-arylbenzofurans derivatives, suggesting organ-specificity in wax defense mechanisms. Leaves exhibit significantly higher protein content than other parts, serving dual roles in photosynthesis and nutrient storage. In addition, seeds primarily store energy as GM polysaccharides, supplemented by xylan, arabinan, and trace alkaloids, enabling efficient energy reserves and seed-specific defense mechanisms (Table 2).

### 3.4. Environmental Modulation of Metabolite Profiles in S. cannabina

In *S. cannabina*, environmental factors remodel both primary and secondary metabolic networks and thereby affect the species, proportions, and fine chemical structures of metabolites. Under waterlogging stress conditions, transcriptome analysis of roots and stems revealed significant differential expression of numerous genes associated with carbon metabolism, glycolysis/gluconeogenesis, phenylpropanoid biosynthesis, and other secondary metabolite pathways. This indicated that water excess and hypoxia can systemically reprogram carbon flux allocation and aromatic skeleton synthesis, fundamentally altering the structure and abundance of metabolites such as cell wall polysaccharides and phenolics [3,69]. Salinity stress and soil improvement also exert feedback effects on metabolite chemistry through the “soil–plant” nutrient pathway. Applying biochar and beneficial microorganisms to saline-alkali soils in the Yellow River Delta significantly reduced soil salinity, increased organic carbon and nitrogen, phosphorus, and potassium availability, and promoted *S. cannabina* growth [48]. This indicates that varying salinity and nutrient backgrounds alter plant carbon-nitrogen balance and metabolic precursor supply, thereby influencing the composition and proportion of end-metabolites such as seed storage polysaccharides and lipids. Correspondingly, *S. cannabina* seeds from different sources exhibited significant variations in molecular weight distribution, galactose/galactose molar ratio, and branching degree [70]. Enzymatically hydrolyzed degradation products of varying molecular weights also exhibited significant differences in antioxidant and immunomodulatory activities [71]. This indicates that environmentally driven metabolic reprogramming can ultimately be recorded by altering the fine chemical structure and biological properties of key metabolites such as polysaccharides. Therefore, when comparing the chemical composition profiles and content levels of *S. cannabina* reported by different research institutes, it is essential to fully account for the influence of environmental factors such as moisture, salinity, and soil nutrients on the chemical characteristics of metabolites.

### 3.5. Component Separation and Detection Methods

#### 3.5.1. Extraction Strategy and Solvent System Selection

The chemical properties of different parts of *S. cannabina* determine the tailored selection of extraction solvents and methods. The primary component in seeds is GM, commonly extracted via hot water extraction combined with ethanol precipitation. Specifically, hot water extraction was conducted at a solid-to-liquid ratio of 1:50 (*w*/*v*), with the resulting extract undergoing ethanol precipitation to yield native GM. To further obtain fractions with varying degrees of polymerization, the residual extract can be hydrolyzed using β-mannanase (20 U/g GM). The hydrolysate was then subjected to fractionated precipitation using 40%, 50%, and 65% ethanol solutions, yielding three fractions: GM40, GM50, and GM65 [6]. Furthermore, GM oligosaccharides were prepared directly from seed powder via enzymatic hydrolysis. The optimal process conditions were pH 5.0, 50 °C, and 150 r/min for a 72 h reaction. Following completion, centrifugation yielded the supernatant [68]. For the 2-arylbenzofuran derivatives abundant in stems, extraction was typically performed using 85% ethanol at room temperature. This method effectively enriches the target compounds [64]. *S. cannabina* powder (S. C. powder), as a byproduct of SG, is typically extracted using 95% ethanol warm maceration. The macerate was concentrated under reduced pressure to yield an ethanol extract, serving as the starting material for subsequent separation [61,65]. The extraction of leaf proteins was performed using the salt-soluble method. After optimization via orthogonal design, the optimal process parameters were determined as follows: solid-to-liquid ratio of 1:20 (g/mL), pH 8, NaCl concentration of 2%, extraction temperature of 60 °C, and extraction time of 120 min [72]. For polyphenolic compounds in *Sesbania* Natto, the optimal extraction conditions obtained through response surface optimization using ethanol solvent extraction were: ethanol concentration 30%, extraction time 80 min, extraction temperature 61 °C, and solid-to-liquid ratio 1:15 (g/mL) [73]. Additionally, for the analysis of wax composition in the cuticle layer of stems and leaves, organic solvent extraction is required. This involved vortexing chloroform extraction at room temperature on stem and leaf samples of known surface area (30 s per extraction, repeated three times). The pooled extract was then nitrogen-blown dry before undergoing derivatization for subsequent analysis [67].

#### 3.5.2. Separation and Enrichment Process

Following initial extraction, the components must undergo stepwise separation and enrichment based on their polarity differences. After concentration, the ethanol extract from the stems enriched the moderately polar 2-arylbenzofuran derivatives primarily in the ethyl acetate fraction through sequential liquid–liquid partitioning using petroleum ether and ethyl acetate [64]. The alcohol extract of S. C. powder underwent precise fractionation through systematic acid-base solvent extraction. The extract was first dissolved in a 2% HCl solution, and the acidic aqueous layer was chloroform-extracted to remove acidic impurities. Subsequently, the remaining aqueous phase was adjusted to pH 10 with Na_2_CO_3_, and the basic fraction (Residue I) was extracted with chloroform. The aqueous phase was then extracted with water-saturated n-butanol to obtain the moderately polar fraction (Residue II). This procedure effectively distributed the different components of the extract into distinct solvents [61,65]. For the leaf protein and natto polyphenol extracts, after extraction, the supernatants were collected by centrifugation (6000 rpm for 10 min and 5000 rpm for 10 min) and directly used for subsequent content determination and activity analysis [72,73].

#### 3.5.3. Chromatographic Purification and High-Purity Sample Preparation

To obtain high-purity monomer compounds from the enrichment fraction, further column chromatography or preparative liquid chromatography techniques are required. The ethyl acetate fraction from the stems was initially separated by silica gel column chromatography (petroleum ether-ethyl acetate-methanol gradient elution). The target fractions were further purified by semi-preparative HPLC (C18 column, acetonitrile-water system), yielding high-purity monomers including sesbcanfuran A (8 mg) and sesbcanfuran B (5 mg) [64]. Based on obtaining two crude extracts (Residue I and Residue II) through solvent extraction from S. C. powder, two high-purity monomers were successfully isolated via systematic silica gel column chromatography. First, elution with cyclohexane-chloroform (4:1) was employed, and the fraction was sublimation-purified to afford white needle-like crystals of 2-Hydroxy-3-methyl-γ-pyrone. Subsequently, the elution system was changed to methanol-chloroform-concentrated ammonia solution (5:5:1). After washing the subsequent eluent with methanol and recrystallizing from water, another white needle-like crystal was 6-armino-9-β-D-ribofuranosyl purine [61].

#### 3.5.4. Structural Identification Methods

The structural identification of chemical components in *S. cannabina* requires the integrated application of multiple spectroscopic techniques. Table 3 systematically summarizes the core identification methods, key instrument parameters, and characteristic identification information employed for target components from different plant parts, providing a reference for the rapid identification and structural elucidation of similar compounds.

#### 3.5.5. Quantitative Analysis Methods

To accurately evaluate the resource quality and functional activity of *S. cannabina*, researchers have established multiple reliable quantitative methods for its various components (including chromatography, colorimetry, and gravimetric analysis). Table 4 systematically summarizes relevant analytical conditions and literature data, providing methodological references for quality control and data comparison.

## 4. Pharmacological Effects

### 4.1. Traditional Uses—Medicinal Records in Folk or Local Literature

According to traditional medicinal records, *S. cannabina* is referred to as “sky-facing centipede”. Its roots, leaves, and seeds are all employed for medicinal applications. As documented in the *New Compilation of Quanzhou Materia Medica*, the roots and leaves are described as having a sweet and slightly bitter flavor, a neutral nature, and an affinity for the heart, kidney, and bladder meridians. It exhibits pharmacological properties that are described in Traditional Chinese Medicine (TCM) terminology as clearing heat and promoting diuresis (indicating anti-inflammatory and diuretic effects), as well as cooling blood and detoxification (suggesting hemostatic and anti-infective activities) [76]. According to the 1979 *Fujian Medicinal Flora*, *S. cannabina* was systematically described regarding its morphology, distribution, and medicinal applications. The text records it as a shrubby legume found in fields and moist roadsides, widely cultivated in Fujian’s coastal areas. The medicinal part is the whole plant, which is typically harvested during summer and autumn for use either in fresh or dried form [77]. Within the framework of TCM, *S. cannabina* is characterized by a sweet and slightly bitter taste and a neutral property, exhibiting anti-inflammatory, antibacterial, and antioxidant effects. Accordingly, its roots are used clinically to treat conditions such as diabetes mellitus, impotence, nocturnal emissions, leukorrhea, and uterine prolapse. The leaves are indicated for surgical disorders, including acute conjunctivitis (pink eye) and venomous snake bites. Common preparations include oral decoctions or the expressed juice of fresh leaves for internal use, while the crushed residue is applied topically to wounds [77]. Furthermore, this book also documents several folk empirical prescriptions. For example, it records “decocting *S. cannabina* roots with pig bladder in water for the treatment of diabetes mellitus”. This prescription demonstrates the significant value of *S. cannabina.* According to the *National Compilation of Chinese Herbal Medicine*, the roots and leaves of *S. cannabina* have widespread folk medicinal applications. They are noted for treating conditions such as hemorrhoids, hives, abscesses, skin infections, and traumatic bleeding. Primary methods include oral decoction or topical application as a poultice, with therapeutic effects aimed at removing blood stasis and promoting tissue regeneration [78]. In the 1999 edition of the *Chinese Materia Medica*, *S. cannabina* is medicinally known as “Skyward Centipede.” Its primary functions are summarized as anti-inflammatory, diuretic, hemostatic, and anti-infective, effects, with indications including hemorrhoids, abscesses, sores, and traumatic bleeding [79].

Ethnobotanical literature documents consistent traditional uses of *S. cannabina* across different areas. For instance, in South Asian traditional medicine systems (e.g., India and Bangladesh), a seed powder is typically prepared by mixing with flour for the external treatment of tinea and skin wounds [80]. In Malaysia and Indonesia, the leaves and seeds of *S. cannabina* are utilized for managing skin conditions and enhancing wound repair, leveraging their purported anti-inflammatory and detoxifying activities [81].

The medicinal value of *S. cannabina* has been systematically documented in traditional Chinese medical texts. These records consistently emphasize its functions in clearing heat, detoxification, promoting diuresis, cooling blood, stopping bleeding, and treating external injuries. Ethnobotanical studies in Southeast Asia further report its use for managing skin diseases, wound healing, and inflammation. This pharmacological convergence across different traditional medicine systems underscores its potential therapeutic relevance and provides a valuable foundation for mechanistic investigation and drug development.

### 4.2. Pharmacological Activity Research

To provide an overview of the reported pharmacological activities of *S. cannabina*, the major bioactive effects identified in current research—including antioxidant, anti-inflammatory, immunomodulatory, hypoglycemic, antitumor, neuroprotective activities, and regulation of intestinal flora—are summarized in Figure 3. These diverse functions highlight the broad therapeutic potential of *S. cannabina* and form the basis for the detailed discussions presented in the following subsections.

#### 4.2.1. Antioxidant Effect

GMs and their degradation products isolated from *S. cannabina* have been shown to exhibit significant antioxidant activity. Li et al. reported that GM from *S. cannabina* demonstrated dose-dependent scavenging activities against 2,2-diphenyl-1-picrylhydrazyl (DPPH), 2,2′-azino-bis(3-ethylbenzothiazoline-6-sulfonic acid) (ABTS), and hydroxyl radicals in vitro, in addition to a potent ferrous ion-chelating capacity [82]. Tao et al. investigated the effects of incompletely degraded products (IDPG) from *S. cannabina* seeds using both in vitro and in vivo models. They observed an enhanced antioxidant capacity in RAW264.7 cells and laying hens, manifested by increased superoxide dismutase (SOD) activity and decreased malondialdehyde (MDA) levels [71]. In addition, Selenylation modification of GM further enhances its antioxidant capacity. Tao et al. reported that the selenylated derivative (SeGM) demonstrated greater efficacy in protecting RAW264.7 cells against H_2_O_2_-induced oxidative damage compared to the native polysaccharide [83]. Yan et al. evaluated low-molecular-weight galactomannans (GA-LMW-GM and GMOS) derived from *S. cannabina* seeds, demonstrating their potent antioxidant activity in both in vitro and in vivo models. In free radical scavenging assays, GA-LMW-GM exhibited stronger activity, with IC_50_ values of 1.9 mg/mL for DPPH and 2.8 mg/mL for superoxide anion (O_2_•^−^), compared to GMOS (IC_50_ = 4.9 mg/mL for DPPH and 4.4 mg/mL for O_2_•^−^). In a zebrafish model, BPAF exposure induced significant oxidative stress; however, supplementation with GA-LMW-GM markedly restored antioxidant defenses, increasing SOD activity by 102.35% and reducing MDA levels by 56.73%. Although GMOS also enhanced SOD activity and decreased MDA content, its effect was comparatively weaker than that of GA-LMW-GM [84]. In addition, Polyphenolic extracts derived from fermented *S. cannabina* products demonstrated notable scavenging capacity against DPPH, hydroxyl, and ABTS^+^ radicals, as well as strong ferric-reducing power, indicating their potential utility as natural antioxidants. Collectively, these findings highlight *S. cannabina* as a promising source of natural antioxidant agents [73]. However, most current findings are based on in vitro assays or animal models, which makes it difficult to extrapolate these results to actual therapeutic settings. In vitro studies reflect biochemical effects under controlled conditions but do not account for absorption, metabolism, bioavailability, or systemic toxicity. In vivo studies incorporate greater physiological complexity, yet species differences in metabolism and immune responses restrict their relevance to humans. Importantly, no clinical studies have evaluated the antioxidant effects, pharmacokinetics, or safety of *S. cannabina* in humans. Therefore, despite consistent in vitro and animal evidence supporting its antioxidant potential, its therapeutic significance remains uncertain. Further pharmacokinetic, toxicological, and clinical studies are required before *S. cannabina*-derived antioxidants can be considered for clinical or nutraceutical applications.

#### 4.2.2. Regulating Inflammation

While direct evidence for the anti-inflammatory activity of *S. cannabina* remains limited, emerging studies indicate that its polysaccharide components can modulate key inflammatory signaling pathways. Li et al. reported that GM from *S. cannabina* stimulated nitric oxide (NO) production in RAW264.7 macrophages and up-regulated the expression of inducible nitric oxide synthase (iNOS) and cyclooxygenase-2 (COX-2), suggesting a potential role in modulating macrophage-mediated inflammatory responses. Similarly, Tao et al. examined IDPG of varying molecular weights and observed a significant up-regulation in the secretion of NO, tumor necrosis factor-α (TNF-α), interleukin-6 (IL-6), and TLR4 expression in RAW264.7 cells, indicating the immunomodulatory potential of these degraded polysaccharides from *S. cannabina* [85]. These findings are based mainly on in vitro cell models, which cannot replicate the complexity of immune regulation in living organisms. Although the results suggest that *S. cannabina* polysaccharides may influence inflammatory pathways, no animal studies or clinical evidence have yet validated these effects. Therefore, their therapeutic relevance remains uncertain. Further in vivo research and clinical evaluation are needed to determine their safety, efficacy, and potential application as anti-inflammatory agents.

#### 4.2.3. Immune Regulatory Effect

The immunomodulatory properties of GM and its degradation products from *S. cannabina* have been systematically investigated. In laying hens, Tao et al. demonstrated that dietary supplementation with IDPG significantly elevated serum immunoglobulin levels [71]. Structure-activity relationship analysis revealed that GM and its derivatives (GM40, GM50, GM65, GMOS) enhanced immune activity by promoting macrophage proliferation. Maximal efficacy was observed at polysaccharide concentrations of 400–800 μg/mL, and a positive correlation was identified between the number of hydroxyl groups in the derivatives and their immunostimulatory potency [83]. Collectively, these findings indicate that polysaccharides from *S. cannabina* and their degradation products can modulate immune responses. However, most of the current evidence comes from in vitro assays and non-mammalian models such as poultry, which limits the extrapolation of these results to mammals or humans due to differences in immune complexity and metabolic processes. Moreover, no mammalian in vivo studies or clinical trials have evaluated their immunological efficacy or safety. Thus, further investigations in higher-order animal models and clinical settings are necessary to determine their true immunomodulatory relevance and potential applications.

#### 4.2.4. Regulation of Intestinal Flora

GM from *S. cannabina* seeds and its degradation products contribute to gastrointestinal health by enhancing digestive stability, modulating the gut microbiota, and thereby promoting the production of beneficial metabolites. Zhou et al. reported that GM from *S. cannabina* seeds exhibits significant structural stability, resisting hydrolysis by gastrointestinal enzymes in vitro. This stability allows it to reach the colon intact, where it serves as a microbiota-accessible carbohydrate. Subsequent in vitro simulations of digestion and fecal fermentation demonstrated that GM fractions of different molecular weights (SG40, SG50, SG65) remained undegraded during the upper gastrointestinal phase but were efficiently utilized by gut microbiota during colonic fermentation, with consumption rates of 74.45%, 68.74%, and 85.34% within 24 h, respectively. This microbial fermentation yielded elevated levels of short-chain fatty acids (SCFAs), which modulated the gut microbiota composition by altering the acetate/propionate ratio and regulating the abundance of dominant bacterial taxa [86]. Dietary supplementation with the IDPG in aged laying hens significantly increased cecal propionate concentration, enriched Bacteroidetes, and reduced the Firmicutes/Bacteroidetes ratio. 16S rRNA sequencing revealed that IDPG increased microbial diversity, and functional prediction analyses indicated enhanced pathways for carbohydrate and lipid metabolism, collectively suggesting improved gut microbiota homeostasis and host disease resistance [87]. In addition, utilizing high-throughput sequencing, Li et al. demonstrated that GM ameliorates gut microbiota dysbiosis in diabetic mice. This was achieved by modulating the microbial community structure, such as enriching Bacteroidetes and reducing the Firmicutes/Bacteroidetes ratio, which subsequently improved microbial diversity and metabolic function, ultimately enhancing host metabolic homeostasis [88].

The beneficial effects of *S. cannabina* GM are primarily underpinned by gut microbial fermentation, which generates SCFAs like propionate. These metabolites lower intestinal pH, inhibit pathogens, and fortify the mucosal barrier. Furthermore, GM and its derivatives selectively enrich beneficial bacteria, thereby competitively excluding pathogens. However, it is important to note that the current evidence, largely derived from in vitro and animal models, provides foundational understanding but cannot be directly applied to humans without further validation. No clinical studies have yet evaluated the effects of *S. cannabina* in human populations. Future research should focus on in vivo studies across diverse animal models and utilize metabolomics combined with causal microbiology approaches to establish the key microbe-metabolite interactions responsible for the observed protective effects.

#### 4.2.5. Hypoglycemic Effect

GM from *S. cannabina* seeds demonstrated significant antidiabetic activity in a type 2 diabetic mouse model (Li et al.). Treatment effectively lowered blood glucose, lipid, and urea nitrogen levels, concomitantly improving insulin sensitivity. Oral glucose tolerance tests (OGTT) confirmed enhanced glycemic control, indicating its role in restoring glucose homeostasis [88]. While these promising results were observed in animal models, it is crucial to recognize that clinical studies in humans are currently lacking. The findings in mice, though encouraging, cannot be directly translated to human therapeutic applications without further validation. The absence of clinical evidence makes it necessary to conduct human clinical trials to confirm the safety, efficacy, and appropriate dosing of *S. cannabina* for managing blood glucose levels in humans.

#### 4.2.6. Anti-Tumor Effect

GM from *S. cannabina* seeds and its low-molecular-weight hydrolysates have been confirmed to possess good antitumor activity. Zhou et al. isolated a series of GM fractions and their low-molecular-weight hydrolysates (GM40, GM50, GM65) from *S. cannabina* seeds. In vitro MTT assays revealed that these compounds significantly inhibited the proliferation of several human cancer cell lines (A549, HeLa, HepG2, MCF-7), with GM40 exhibiting the most potent antitumor activity. Immunohistochemical analysis indicated that its mechanism may involve the upregulation of caspase-12 expression, suggesting the induction of apoptotic cell death [6].

Phytochemical investigation of the stems of *S. cannabina* led to the isolation of eight 2-arylbenzofuran derivatives, a class of secondary metabolites with reported antitumor activity. This series included two new compounds, designated sesbcanfuran A (1) and sesbcanfuran B (2), along with six known analogues (3–8), such as sesbagrandiflorain E and spinosan A. The cytotoxicities of these compounds were assessed against three human cancer cell lines (HeLa, MCF-7, A549) via MTT assay. Sesbcanfuran B demonstrated the most potent cytotoxicity, with IC_50_ values of 1.5 μM (MCF-7), 4.8 μM (A549), and 32.1 μM (HeLa). Sesbcanfuran A and the methyl ester derivative showed moderate activity (IC_50_ 22.3–39.2 μM), while the remaining compounds (3, 5–8) were weakly active or inactive (IC_50_ > 40 μM). These results indicate that 2-arylbenzofuran derivatives, particularly sesbcanfuran B, are promising natural lead compounds for anticancer drug development [64].

In conclusion, GM extracted from *S. cannabina* seeds and its hydrolysates and the 2-arylbenzofuran derivatives exhibit notable anticancer potential based on in vitro studies. However, these findings remain preliminary, as no in vivo studies have yet confirmed their antitumor efficacy in animal models, and clinical evidence in humans is entirely absent. This lack of translational data means that the promising in vitro results cannot be directly applied to clinical cancer therapy. Therefore, future research should prioritize in vivo validation of these compounds and conduct well-designed clinical trials to evaluate their safety, pharmacological behavior, and therapeutic potential before they can be considered viable candidates for anticancer drug development.

#### 4.2.7. Neuroprotective Effect

In recent years, the potential neuroprotective effects of low molecular weight galactomannans (LMW-GMs) derived from *S. cannabina* have attracted increasing attention. Yan et al. demonstrated that two GA-LMW-GM and GMOS ameliorated bisphenol AF (BPAF)-induced neurotoxicity in zebrafish. Larvae exposed to BPAF showed impaired locomotor activity (swimming speed) and neuronal expression, both of which were significantly rescued by GA-LMW-GM and GMOS treatment, restoring values by 14.28% and 14.20%, respectively, to near-normal levels. These results indicate the neuroprotective potential of these LMW-GMs against environmental toxin-induced neural damage [84].

In summary, LMW-GMs from *S. cannabina*—particularly GA-LMW-GM—exhibit promising neuroprotective activity in zebrafish models. These findings indicate potential value in neuroprotection. However, current evidence is limited to non-mammalian systems, which cannot fully represent mammalian or human neural physiology. Factors such as metabolism, blood–brain barrier properties, and immune–neural interactions may produce different outcomes in higher organisms. To date, no mammalian studies or clinical trials have examined their neuroprotective efficacy. Thus, although LMW-GMs appear promising, further validation in mammalian models and well-designed clinical studies is required to confirm their effects, elucidate mechanisms, and assess their potential for neuroprotective applications.

#### 4.2.8. Other Potential Bioactivities (Inferred from Extracted Components)

Beyond the well-documented antioxidant, immunomodulatory, and antitumor effects, *S. cannabina* may also possess additional bioactivities that have not been systematically investigated. Mishra et al. (2021) reported that *S. cannabina* seed oil is rich in unsaturated fatty acids, primarily linoleic acid and oleic acid, which are well-established for their cardioprotective properties and ability to improve lipid profiles, implicating its potential value in promoting cardiovascular health [89]. A systematic review and meta-analysis of 40 randomized controlled trials (*n* = 2175) by Wang et al. (2023) demonstrated that dietary linoleic acid supplementation significantly lowers LDL-C, supporting its beneficial role in lipid regulation [90]. A 2024 systematic review synthesizing evidence from prospective cohort studies and clinical trials demonstrated that higher dietary intake of linoleic acid is associated with reductions in total cholesterol and LDL-C, with these effects translating into a significantly lower risk of cardiovascular disease and type 2 diabetes [91]. Collectively, these findings suggest that *S. cannabina* seed oil may hold potential for cardiovascular protection and lipid-lowering applications.

### 4.3. Potential Targets and Signaling Pathways

#### 4.3.1. Activation of the Nrf2/ARE Pathway

GM extracts from *S. cannabina* exhibit significant effects in alleviating oxidative stress, and the underlying mechanism is believed to be closely associated with the activation of the nuclear factor erythroid 2-related factor 2 (Nrf2)/antioxidant response element (ARE) signaling pathway (Figure 4a). It is proposed that this pathway enhances the endogenous defense system by upregulating the activities of key antioxidant enzymes such as SOD and glutathione peroxidase (GSH-Px), while simultaneously reducing intracellular reactive oxygen species (ROS) and MDA levels, thereby mitigating oxidative damage [85].

#### 4.3.2. Activation of the TLR/NF-κB Pathway

GM, a major bioactive component of *S. cannabina,* plays a pivotal role in its immunomodulatory effects. Studies indicate that GM and its derivatives activate Toll-like receptors (TLR2/4), triggering the NF-κB signaling cascade (Figure 4b). This activation upregulates the expression of iNOS, COX-2, and cytokines (e.g., IL-1β, IL-6, TNF-α, IFN-γ), mediating the observed immunostimulatory effects. Comparative analysis of different molecular weight fractions reveals a primary dependence on TLR4, establishing the TLR4/NF-κB pathway as the central mechanism for the immune-enhancing activity of *S. cannabina*, which may also contribute to its anti-inflammatory potential [85].

#### 4.3.3. Caspase-Dependent Apoptotic Signaling

The antitumor potential of *S. cannabina* is closely related to its ability to induce apoptosis in tumor cells. Bioactive fractions have been shown to significantly inhibit proliferation across multiple human cancer cell lines. Mechanistically, this effect may involve the endoplasmic reticulum stress pathway, characterized by specific upregulation of caspase-12 expression, which subsequently activates the caspase cascade, ultimately driving programmed cell death (Figure 4c). These findings provide preliminary pharmacological evidence supporting the development of *S. cannabina*-derived compounds as anticancer therapeutics [6].

#### 4.3.4. Gut Microbiota–Short-Chain Fatty Acid Metabolic Axis

Polysaccharides from *S. cannabina* (e.g., GM, IDPG) serve as non-digestible dietary components that undergo microbial fermentation in the colon. This process enriches beneficial bacteria (e.g., Bacteroidetes), restores microbial diversity, and boosts the production of SCFAs like propionate. As key metabolites, SCFAs function as signaling molecules and energy sources, thereby activating the “gut microbiota–SCFA axis”, which improves intestinal immunity and systemic metabolic homeostasis (Figure 4d). This suggests that the “gut microbiota–SCFA metabolic axis” may represent a critical mechanism by which *S. cannabina* exerts its antidiabetic and metabolic regulatory effects [86].

### 4.4. Toxicology and Safety

#### 4.4.1. Toxic Components

Powell et al. identified sesbanimide in seeds of several toxic *Sesbania* species, linking it to livestock poisoning. It is crucial to note that secondary metabolite profiles differ substantially among *Sesbania* species. Therefore, toxicity findings from one species cannot be extrapolated to the entire genus. Powell’s work suggests the need to screen for potential toxic compounds in *S. cannabina* seeds specifically. Separate studies have conducted toxicity assessments on SGF gum derived from *S. cannabina* seeds, revealing that dietary levels of 5–10% SGF inhibit body weight gain and may exhibit potential genotoxic effects [92].

Ecotoxicological studies indicate that *S. cannabina* exhibits differential tolerance to metals such as Pb, Zn, Cu, and Cd. This suggests potential for heavy metal accumulation when grown in contaminated soils, raising concerns for its safe use in medicinal or food products. Consequently, quality control measures—including monitoring of Pb, Cd, As, and Hg in both soil and plant tissues—are essential to ensure safety [93].

#### 4.4.2. Toxicology and Safety Assessment

Li et al. (1988) [94] conducted toxicity assessments and safety evaluations of SGF gum (a polysaccharide colloid similar to guar gum) derived from *S. cannabina* seeds through acute toxicity, subchronic and multigenerational growth and reproduction studies, teratogenicity, genotoxicity, and short-term human trials. Acute toxicity studies determined an oral LD_50_ in rats of approximately 19 g/kg, indicating very low acute toxicity. At high doses, adverse effects were attributed to mechanical gastric obstruction from gel expansion. Subchronic and multigenerational studies identified a dietary level of 1% SGF as the No Observed Adverse Effect Level (NOAEL), with no significant changes in hematological, biochemical, or histological parameters. At higher levels (5–10% diet), suppressed weight gain was observed. Additionally, teratogenicity studies also reported an increased incidence of sixth sternum ossification defects, suggesting potential impacts on skeletal development [94].

Beyond the safety profile of SGF gum itself, several bioactive constituents identified in *S. cannabina* may also present potential toxicological risks, as suggested by studies on structurally related compounds in other plant species. Triterpenoid saponins, which are present in *S. cannabina*, are widely recognized for their hemolytic properties through interactions with membrane sterols, and high doses have been associated with gastrointestinal irritation and liver or kidney injury in experimental animals [95,96]. Phenolic compounds and tannins, although known for their antioxidant activity, can precipitate dietary proteins, reduce protein digestibility, inhibit digestive enzymes, and induce gastrointestinal discomfort when consumed at high levels [97]. For the 2-arylbenzofuran derivatives identified in *S. cannabina*, current evidence is largely limited to their in vitro activity against cancer cell lines, while data regarding their effects on normal cells or whole organisms remain scarce. Studies on structurally related benzofuran compounds have shown dose-dependent responses in normal endothelial cells [98]. These findings indicate that this chemical class may possess non-target biological activities that warrant further investigation. Therefore, additional studies are needed to clarify their in vivo safety profiles and toxicological characteristics. Similarly, purine nucleosides such as adenosine—another bioactive component of *S. cannabina*—may induce cardiovascular adverse effects, including hypotension, atrioventricular block, and arrhythmias when administered at high doses in clinical settings [99], further underscoring the importance of comprehensive safety assessment for these constituents.

Collectively, these findings indicate that while *S. cannabina* contains multiple pharmacologically valuable constituents, their potential adverse effects warrant careful toxicological evaluation. Current evidence remains limited to polysaccharides and a few isolated compounds, with no comprehensive toxicokinetic, long-term toxicity, or clinical studies available. Further work is needed to clarify the toxicity mechanisms, dose thresholds, metabolic pathways, and safety margins of key active components, thereby supporting the safe development and application of *S. cannabina*-derived products.

## 5. Current Research Status and Existing Challenges

While initial efforts have established a foundation by collecting *S. cannabina* germplasm and characterizing its genetic diversity and ecological adaptability, the resource conservation and evaluation system remains underdeveloped, leading to significant genetic erosion. Furthermore, genomic insights are primarily restricted to analyses of a few gene families without systematic depth or extensive functional validation (e.g., via gene knockout/overexpression). This gap ultimately limits the translation of research findings into genetic improvement and molecular breeding applications.

Although initial phytochemical investigations have identified *S. cannabina* as a source of diverse bioactive compounds (e.g., alkaloids, polysaccharides, phenylpropanoids), research remains largely preliminary. A systematic characterization is lacking, particularly regarding the quantitative profile, spatial distribution across tissues, and temporal accumulation patterns during development or under stress. The chemical complexity and heterogeneous distribution of constituents also complicate extraction and purification, highlighting the need for efficient, scalable methods. In future studies, the application of advanced techniques like iEESI-MS is anticipated to accelerate the rapid separation and identification of bioactive molecules, thus enabling a more comprehensive phytochemical elucidation [100].

While traditional uses and preliminary pharmacological data suggest that *S. cannabina* and its seeds possess heat-clearing, detoxifying, and hepatosplenic benefits, along with potential cardioprotective and neuroprotective effects, robust clinical evidence is lacking. Current research relies predominantly on in vitro studies and small-scale animal models, underscoring the need for large-sample clinical trials and epidemiological studies to confirm efficacy and safety across diverse populations, especially in the elderly.

A critical gap remains in understanding the mechanistic link between the bioactive constituents of *S. cannabina* (e.g., flavonoids, polysaccharides, saponins) and their observed pharmacological effects. Current studies often confirm bioactivity at the extract level but fail to identify the key active components or elucidate their modes of action—such as modulation of specific signaling pathways, gene regulation, or interactions with molecular targets in vivo. This lack of mechanistic insight severely limits its potential for precise development as a therapeutic or nutraceutical.

## 6. Development Prospects and Application Recommendations

As a leguminous plant with multiple applications, the future development of *S. cannabina* requires a foundation of systematic research efforts and integrated industrial development. First, a comprehensive evaluation of germplasm resources and the development of core germplasm should be prioritized. The team led by Academician Xiaofeng Cao has established the largest *S. cannabina* germplasm repository in China, collecting 546 accessions, and has bred a salt-tolerant, high-yielding new variety, “Zhongkejing NO. 1”. However, further efforts are needed to construct a core germplasm collection through whole-genome resequencing and SPO sampling, thereby strengthening genetic diversity analysis and facilitating molecular breeding applications. Second, it is necessary to integrate multi-omics approaches with modern isolation techniques to explore bioactive constituents. While current studies confirm a rich diversity of active compounds, their spatiotemporal distribution and accumulation dynamics remain poorly understood. Combining metabolomics and transcriptomics can help map these constituents, while the development of green technologies, such as ultrasound-assisted extraction, can enable targeted preparation. Third, efforts should be made to advance pharmacology research toward preclinical translation. Although existing evidence supports the antioxidant efficacy of *S. cannabina*, future research should focus on transitioning from crude extract testing to mechanistic investigations, alongside standardized assessments of dose–response relationships, pharmacokinetics, and safety to establish a foundation for clinical trials. Fourth, the development of high-value-added functional products from *S. cannabina* warrants emphasis. Capitalizing on its high crude protein content (>16% in the whole plant) and substantial biomass yield (fresh forage yield of 4.29 kg/m^2^), the production of forage-derived commodities like grass meal and pellets can be expanded. Furthermore, the creation of functional foods or nutraceuticals tailored for specific populations, such as diabetic individuals, represents a promising direction. These endeavors highlight the necessity of integrated value-chain development, fostering the translation of *S. cannabina* from basic research to industrial application through a coordinated system of production, processing, and utilization. Finally, it is essential to strengthen domestic and international resource sharing and collaborative research. Expanding germplasm exchange with institutions such as ICRISAT and forming interdisciplinary teams integrating botany, chemistry, and pharmacology will help address key bottlenecks in elucidating bioactive mechanisms and developing practical applications. Concurrently, implementing composite cropping systems (e.g., grass-light complementary cultivation, wheat-*Sesbania* rotation) in key ecological zones like Xinjiang and Heilongjiang will synergistically advance saline-alkali land remediation and foster associated industries.

## Figures and Tables

**Figure 1 ijms-26-12129-f001:**
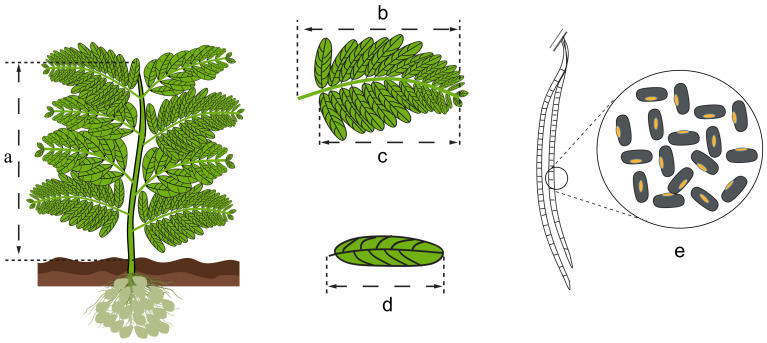
Schematic diagram of *Sesbania*: (**a**) Plant Height, (**b**) Stem, (**c**) Number of Leaflets, (**d**) Leaflet Size, (**e**) Seed Characteristics. Created with BioGDP.com.

**Figure 2 ijms-26-12129-f002:**
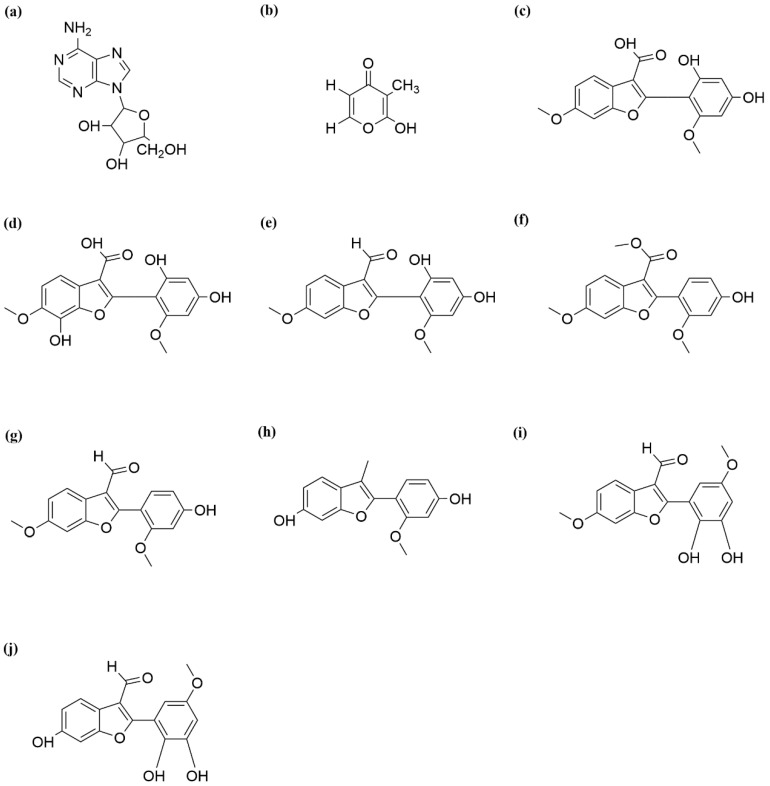
Chemical structures in *S. cannabina*: (**a**) 6-Armino-9-β-D-ribofuranosyl purine, (**b**) 2-Hydroxy-3-methyl-γ-pyrone, (**c**) sesbcanfuran A, (**d**) sesbcanfuran B, (**e**) sesbagrandiflorain E, (**f**) 2-(4-hydroxy-2-methoxyphenyl)-6-methoxy-3-benzuofur ancarboxylic acid methyl ester, (**g**) spinosan A, (**h**) 2-(2′-methoxy, 4′-hydroxy)-aryl-3-methyl-6-hydroxy-benzuofuran, (**i**) sesbagrandiflorain A, (**j**) sesbagrandiflorain B.

**Figure 3 ijms-26-12129-f003:**
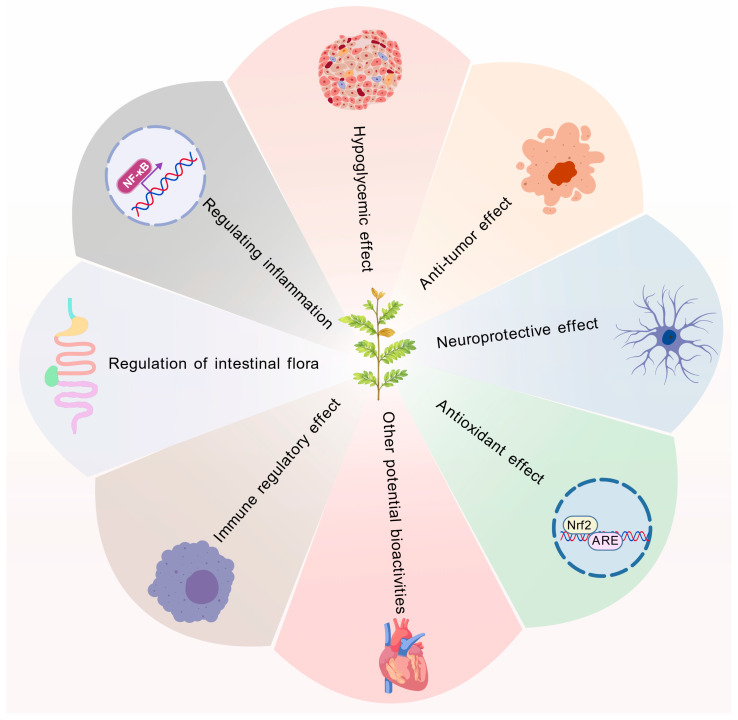
This diagram illustrates the diverse pharmacological activities of *S. cannabina*. Created with BioGDP.com.

**Figure 4 ijms-26-12129-f004:**
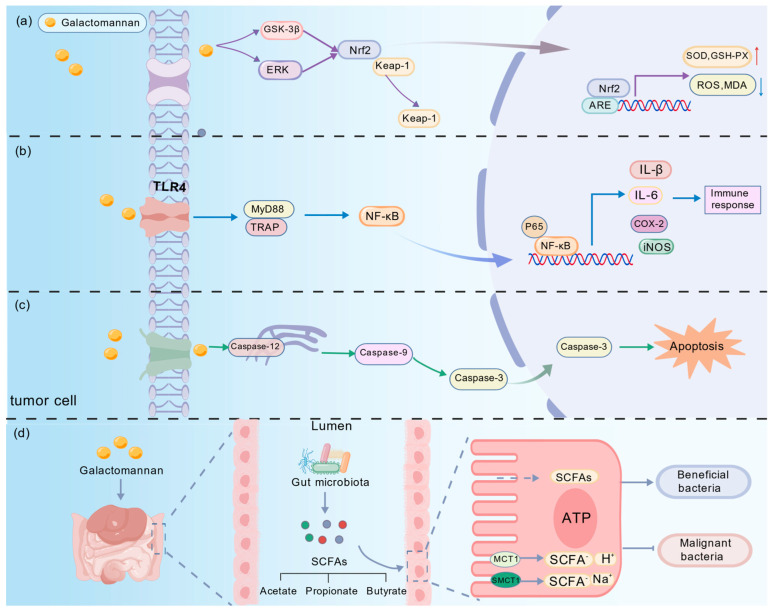
The schematic diagram illustrates the molecular mechanisms underlying the pleiotropic biological effects of GM from *S. cannabina.* (**a**) Antioxidant pathway: GM is suggested to activate the Nrf2/ARE signaling cascade, possibly via ERK and GSK-3β, leading to enhanced expression of antioxidant enzymes (e.g., SOD, GSH-Px) and reduced oxidative stress markers (ROS, MDA). (**b**) Immune regulatory pathway: GM and its derivatives may engage TLR4 and trigger MyD88/TRAF-mediated NF-κB activation, promoting the expression of iNOS, COX-2, and pro-inflammatory cytokines (IL-1β, IL-6), thereby modulating immune responses. (**c**) Antitumor pathway: GM could induce ER stress, leading to caspase-12 activation and subsequent caspase cascade (caspase-9, caspase-3), which may result in apoptosis of tumor cells. (**d**) Gut microbiota–SCFA axis: Non-digestible GM reaches the colon and undergoes microbial fermentation to produce SCFAs (acetate, propionate, butyrate). SCFAs regulate gut homeostasis by increasing beneficial bacteria and suppressing pathogenic bacteria. Created with BioGDP.com.

**Table 1 ijms-26-12129-t001:** Distribution and Traits of Different *Sesbania* Species [12].

Biological Species	*Sesbania cannabina*	*Sesbania bispinosa*	*Sesbania sesban*	*Sesbania javanica*	*Sesbania grandiflora*
Distribution	Iraq, India, Indochina Peninsula, Malaysia, Papua New Guinea, New Caledonia, Australia, Ghana, Mauritania; Hainan, Jiangsu, Zhejiang, Jiangxi, Fujian, Guangxi, Yunnan	Iran, Pakistan, India, Sri Lanka, the Indochinese Peninsula, the Malay Peninsula; Guangdong, Guangxi, Yunnan, and Sichuan (southwestern regions)	Tropical Africa, Southern Africa, the Arabian Peninsula, and the Indian subcontinent; cultivated in Taiwan (Taipei, Changhua, Penghu) and Hainan	Australia, Indonesia, Malaysia, Bangladesh, Myanmar, Thailand, Cambodia, Laos, Vietnam; Taiwan, Hainan (cultivated)	Pakistan, India, Bangladesh, Indochina Peninsula, Philippines, Mauritius; Taiwan, Guangdong, Guangxi, Yunnan
Plant Height (a)	2–3.5 m	1–3 m	2–4 m	2–4 m	4–10 m
Stem (b)	Green, sometimes tinged with reddish-brown, slightly covered with white powder	Rachises and peduncles sparsely covered with small prickles	Pubescent when young, becoming glabrous later; nodes prominently swollen	Pith white, young branches sparsely pubescent	With distinct leaf scars and stipule scars
Number of Leaflets (c)	20–40 pairs	20–40 pairs	10–20 pairs	10–30 pairs	10–30 pairs
Leaflet Size (d)	0.8–4 cm in length, 2.5–7 mm in width	1–1.6 cm in length, 2–3 mm in width	1.3–2.5 cm in length, 3–4 (−6) mm in width	10–40 mm in length, 2–7 mm in width	2–5 cm in length, 0.8–1.6 cm in width
Seed Characteristics (e)	Greenish-brown, glossy, short cylindrical, hilum orbicular, slightly eccentric	Subcylindrical, hilum orbicular, located at the middle	Subcylindrical, slightly compressed, hilum orbicular, concave	Greenish-brown to dark brown, glossy, subglobose	Reddish-brown, slightly glossy, elliptic to subreniform, swollen and slightly compressed, hilum orbicular, slightly concave

**Table 2 ijms-26-12129-t002:** Comparative Analysis of Components in Root, Stem, Leaf, and Seed.

Plant Part	Chemical Composition Category	Primary Compounds	Approximate Content/Characteristics
Root	Fibrous substances	Neutral detergent fiber (NDF), acid detergent fiber (ADF), and acid detergent lignin (ADL)	relatively high content [66]
Stem	Stratum corneum wax	Fatty acids, primary alcohols, aldehydes, alkanes, alkyl esters, diols, terpenes, and sterols (eight compound classes in total), with primary alcohols as the predominant constituents	The total stem wax content was 6.45 μg cm^−2^, with primary alcohols accounting for 30.12% of the total stem wax content [67]
		Identified diol compounds (1,18-tridecanediol, 1,16-tridecanediol) [67]	
	2-Arylbenzofuran derivatives	sesbcanfuran A sesbcanfuran B sesbagrandiflorain E 2-(4-hydroxy-2-methoxyphenyl)-6-methoxy-3-benzuofur ancarboxylic acid methyl ester spinosan A 2-(2′-methoxy, 4′-hydroxy)-aryl-3-methyl-6-hy droxy-benzuofuran sesbagrandiflorain A sesbagrandiflorain B	
Leaf	Protein	High in protein content [66]	
	Stratum corneum wax	Fatty acids, primary alcohols, aldehydes, alkanes, alkyl esters, diols, terpenes, and sterols (8 compound categories total), with primary alcohols being the predominant component	Primary alcohols account for 71.21% of the total leaf wax content [67]
Seed	Polysaccharides (primary source of *Sesbania* Gum)	Galactomannan	Composed of D-galactose and D-mannose [60]
		Xylan and Arabinan [6]	
	Monosaccharide (Seed hydrolysate)	Mannose, Galactose, Glucose, and Xylose	Determined by HPLC-PMP method [68]
	Alkaloids	2-Hydroxy-3-methyl-γ-pyrone and 6-armino-9-β-D-ribofuranosyl purine	

**Table 3 ijms-26-12129-t003:** Structural Characterization of the Chemical Components in *S. cannabina*.

Part	Target Components	Forensic Technology	Key Parameters	Results	References
Seed	Galactomannan	FT-IR	KBr Tablet	815 cm^−1^ (α-Galp), 870 cm^−1^ (β-Manp)	[6]
Seed	Galactomannan	^1^H/^13^C-NMR, HSQC	D_2_O, 600 MHz	δH 5.02 ppm (α-Gal H-1); The main chain consists of β-1,4-mannose, with α-1,6-galactose side chains randomly attached at the C-6 position	[6]
Stem	2-Arylbenzofuran derivatives	HR-ESI-MS	Negative Ion Mode	[M–H]^−^ 329.0064 (sesbcanfuran A); 345.0618 (sesbcanfuran B)	[64]
Stem	2-Arylbenzofuran derivatives	1D/2D-NMR	400 MHz (^1^H), 100 MHz (^13^C)	HMBC confirms the 7-OH position (H-5/C-7 and related)	[64]
S. C. powder	2-Hydroxy-3-methyl-γ-pyrone	EI-MS, EA, UV, IR, ^1^H-NMR	Mr 126; UV 276 nm; IR 1660, 1620 cm^−1^	New pyranone compound	[61]
S. C. powder	6-Armino-9-β-D-ribofuranosyl purine	EI-MS, EA, UV, IR, ^1^H-NMR, [α]	Mr 267; UV 258 nm; IR 1665, 1605 cm^−1^	The structure was confirmed as adenosine, isolated for the first time from the genus *Sesbania*.	[61]
Leaf	Protein secondary structure	FT-IR	Amide Zone I, Amide Zone II	The area under the β-sheet peak is 0.37	[66]
Stem/Leaf	Stratum corneum wax	GC-MS	DM-5 chromatography column; Programmed temperature	A total of 8 classes of compounds were identified; 1,18-tridecanediol, 1,16-tridecanediol were identified in the stems	[67]

**Table 4 ijms-26-12129-t004:** Quantitative Analysis Methods of the Chemical Components in *S. cannabina*.

Part	Analysis Items	Quantitative Methods	Test Conditions	Linearity/Detection Limit	Results	References
Seed	Galactomannan content	HPAEC-PAD	Detection of monosaccharides after acid hydrolysis	—	95.07% (GalM purity)	[6]
Seed	Galactomannan molecular weight	HPGPC-RI	Waters Ultrahydrogel Series Columns, Pure Water	—	M_w 1.42 × 10^6^ Da (GalM)	[6]
Seed	Monosaccharide composition	PMP pre-column derivatization RP-HPLC-UV	C18 column; acetonitrile-phosphate buffer (pH 6.5) gradient elution; detection wavelength 250 nm	5–2500 mg/L, R^2^ > 0.9994, LOD 0.455–1.224 mg/L	Man:Glc:Gal:Xyl = 1:0.44:1.92:0.53 (Molar ratio)	[68]
Seed	Galactomannan molecular weight	GPC-RI	Waters μ-Bondagel Series Columns, Pure Water	—	M_w 2.3 × 10^5^ (SG)	[74]
S. C. powder	Total alkaloids, total flavonoids, total saponins	Weight method	—	—	Total alkaloids: 1.64% Total flavonoids: 0.40% Total saponins: 2.28%	[65]
S. C. powder	Tannin	Coordinate titration method	—	—	1.86% DW	[65]
Leaf	Total Protein	C.I. 250,000	BSA Standard, measured at 595 nm	0.01–0.1 mg/mL, R^2^ = 0.9914	40.02 mg/g (DW)	[72]
Natto	Total polyphenols	Folin–Ciocalteu	Gallic acid standard, measured at 765 nm	10–50 μg/mL, R^2^ = 0.9995	119.24 mg/g (DW)	[73]
Stem	Lignin content	Hydrochloric acid-resorcinol semi-quantitative colorimetric method	Cross-section staining in the field	—	Color intensity is used for quick comparison	[75]
Stem/Leaf	Wax content and composition of the stratum corneum	GC-FID/GC-MS	Chloroform extraction, BSTFA derivatization, DM-5 column temperature programming analysis	—	Total stem wax content: 6.45 μg cm^−2^ (primary alcohols accounted for 30.12%); Total leaf wax content: 15.3 μg cm^−2^ (primary alcohols accounted for 71.21%)	[67]

## Data Availability

No new data were created or analyzed in this study. Data sharing is not applicable to this article.

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
