# Peer review of "Panoramic Research on *Sesbania cannabina*: Germplasm Resources, Phytochemical Constituents, Biological Activities, and Applications"

_ijms, 2025, doi:10.3390/ijms262412129_

Round 1
Reviewer 1 Report
Comments and Suggestions for Authors
I find the proposed review interesting and relevant, as it provides information that may be useful for future research. However, I believe the quality of the manuscript could be further improved by addressing the following points:
Q1. In the introduction, several benefits of S. cannabina are mentioned, but no quantitative data are provided. Including specific numerical values or examples would strengthen the scientific context and support the relevance of the species.
Q2. Similarly, the introduction could better emphasize the novelty and contribution of this review.
Q3. Please indicate which flavonoids have been reported in S. cannabina and include a figure illustrating the chemical structures of the main or most abundant compounds.
Q4. Line 442: Consider dividing the section describing the isolation and purification methods into clear subsections to improve readability.
Q5. Line 466: Likewise, separate the identification and quantitative analysis methods. Presenting them in a table format may enhance clarity and comparison.
Q6. Incorporate a figure summarizing the different pharmacological effects associated with S. cannabina to provide a comprehensive visual overview.
Reviewer 2 Report
Comments and Suggestions for Authors
Dear Authors,
The study addresses a relevant subject; however, the manuscript requires important revisions to improve clarity, methodological detail, and scientific depth. Please find below specific suggestions to strengthen the quality and impact of your work.
Introduction
The introduction adequately presents the general context of the topic; however, the objective of the review is still not sufficiently explicit. I recommend that the authors more clearly specify:
- which scientific questions this review aims to answer, and
- what is the original contribution of this work compared to existing reviews on Sesbania spp.
In addition, I suggest clearly highlighting the knowledge gaps that remain unresolved and justify the relevance of this review.
Materials and Methods
The authors do not describe the criteria used for selecting the included literature. The absence of this information compromises the transparency and reproducibility of the review. I request that a methodological section be included detailing:
• the databases consulted,
• search strategies (keywords/descriptors),
• the time range covered by the search,
• inclusion and exclusion criteria for studies, and
• the total number of articles identified and selected.
Furthermore, please clarify whether this review follows a narrative, scoping, or systematic approach, as this distinction directly affects the interpretation of the robustness of the evidence presented.
Germplasm and Genetics
This section demonstrates a strong understanding of the topic; however, there is an excess of morphological detail. I suggest synthesizing part of the content, prioritizing the information essential to the focus of the review, and considering relocating overly extensive tables to supplementary material in order to improve the flow of the text.
Chemical Compounds and Metabolism
I recommend expanding the discussion on metabolic variability of the species, addressing how environmental factors (such as soil, climate, and phenological stage) influence the chemical profile of metabolites. This discussion would contribute to a better understanding of the divergences among the studies cited.
Pharmacological Activities
The results of in vitro studies are presented in a descriptive manner, without considering the limitations regarding extrapolation to therapeutic applications. I suggest reinforcing the distinction between in vitro and in vivo evidence, as well as the current absence of clinical studies.
It is also essential to include a more robust discussion on safety and toxicity, including possible adverse effects associated with the identified bioactive compounds, since this is crucial for assessing the real pharmacological potential of the species.
Reviewer 3 Report
Comments and Suggestions for Authors
The review paper submitted for review concerns a summary of information on the plant Sesbania cannabina. It covers genetic resources, chemical composition, application and biological activity. The paper is well organised and edited. The diagram showing the molecular mechanisms responsible for the biological activity of the raw material is noteworthy.
While reading, I noticed several shortcomings that should be corrected.
The authors use the unit ‘per mu’. This should be explained and SI units appropriate for scientific work should be used (line 120).
In the section devoted to the chemical composition of the plant in terms of flavonoid, saponin and other secondary metabolite content, the authors refer to a single work from 1992 (reference number 65). Finding more recent publications would certainly increase the value of the work.
The abbreviation GM appears for the first time in section 3.4.1. This abbreviation should be explained.
In the section devoted to the traditional uses of Sesbania cannabina, statements such as ‘clearing heat’ or ‘cooling blood’ are used. Their more contemporary meaning should be explained.
